 SciPost Phys. Lect. Notes 89 (2024)

# Population genetics: An introduction for physicists

**Andrea Iglesias-Ramas[1][⋆][∘], Samuele Pio Lipani[2][†][∘] and Rosalind J. Allen[3,4][‡]**

**1** Institut Curie, PSL Research University, Sorbonne Université, CNRS UMR 168,
Laboratoire Physique des Cellules et Cancer, 75005 Paris, France
**2** Institut Curie, PSL Research University, Sorbonne Université, CNRS UMR3664
and UMR 168, Laboratoire Dynamique du Noyau, 75005 Paris, France
**3** Theoretical Microbial Ecology Group, Institute of Microbiology,
Faculty of Biological Sciences, Friedrich-Schiller University, Jena, Germany
**4** Cluster of Excellence Balance of the Microverse,
Friedrich Schiller University, Jena, Germany

⋆ andrea.iglesias-ramas@curie.fr , † samuele.lipani@curie.fr , ‡ rosalind.allen@uni-jena.de

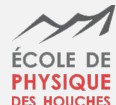

*Part of the 2023-07: Theoretical Biological Physics 2023 collection*
*Session 121 of the Les Houches School, July 2023*
*published in the Les Houches Summer School Lecture Notes series*

## Abstract

**Population genetics lies at the heart of evolutionary theory. This topic forms part of many biological science curricula but is rarely taught to physics students. Since physicists are becoming increasingly interested in biological evolution, we aim to provide a brief introduction to population genetics, written for physicists. We start with two background chapters: chapter 1 provides a brief historical introduction to the topic, while chapter 2 provides some essential biological background. We begin our main content with chapter 3 which discusses the key concepts behind Darwinian natural selection and Mendelian inheritance. Chapter 4 covers the basics of how variation is maintained in populations, while chapter 5 discusses mutation and selection. In chapter 6 we discuss stochastic effects in population genetics using the Wright-Fisher model as our example, and finally we offer concluding thoughts and references to textbooks in chapter 7.**

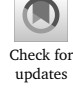

## Contents

∘ These authors contributed equally to the development of this work.

# 1  A brief historical introduction

The diversity of the living world naturally inspires attempts to rationalize the origin and maintenance of different living species. This is the main aim of evolutionary theory. We start these notes with a very brief (and by no means complete) overview of the long history of thought and debate upon which modern evolutionary theory is based. We note that our overview focuses exclusively on the Western world; the relationship between Eastern and Western views of evolution is very interesting [1] but is beyond the scope of these notes.

We start with the question "what is there?". This question was already addressed by Aristotle (384-322 BCE), who made an early attempt to classify the natural world into a hierarchy, or "ladder of life" [2]. Much later, Carl Linnaeus (1707-1778), the "father of modern taxonomy" re-addressed the same question, developing the multi-level taxonomic classification system (from kingdom down to species level) that is still in use today [3].

Classification of species naturally leads to a further question: "how does what is there change in time?". While the traditional view had been that the biological characteristics of species do not change in time (the "static" view of nature), around the turn of the 19th century an opposing "dynamic" view emerged, which proposed that new species can arise, while old species can become extinct. An influential proponent of this idea was Georges Cuvier (1769-1832), the "father of paleontology",[1] who concluded that ancient species had become extinct by comparing living animals with fossils [5].

---

[1]It is important to know that Georges Cuvier also contributed to the foundation of scientific racism [4].

From the dynamic view of nature follows, of course, the question "what is the mechanism that causes species to change in time?". Cuvier suggested that change occurs in a succession of devastating cycles of global extinction, caused for example by floods, alternating with periods of creation of new life forms [5]. The concept of *evolution* emerged as a contrasting, more gradual, hypothesis, in which the heritable characteristics of biological species change in small increments over successive generations, governed by natural laws.

Jean-Baptiste Lamarck (1744-1829) proposed the first theory of evolution [6]. He suggested that evolution is driven by two "forces": a complexifying force that drives organisms' body plans from simple to more complex forms (e.g. jellyfish to vertebrates) and an adaptive force that causes organisms with a given body plan to adapt to their environment (hence, different jellyfish inhabit different parts of the ocean). Famously, Lamarck also believed that characteristics acquired during an organism's lifetime can be inherited by their offspring ("soft inheritance").

The theory of natural selection, proposed in 1858/9 by Charles Darwin (1809-1882) [7] and Alfred Russel Wallace (1823-1913) [8], set out a different hypothesis, in which the characteristics of individuals within a population are variable, heritable, and linked to differential reproduction. The theory of natural selection caused a great deal of debate at the time of its introduction; it became widely accepted only in the early-mid 20th century, long after the death of Darwin. Natural selection (and some of the early reasons for controversy) will be discussed in more detail below. Population genetics, the topic of these notes, is the theory of natural selection applied to populations.

## 2 Essential biological background and terminology

In this section, we review some key biological concepts and terms that play an important role in the rest of these notes. It is important to note, however, that the basis of population genetic theory was developed without the molecular-level knowledge that we present here. For example, DNA was discovered to be the unit of inheritance only in 1944 [9], 36 years after the establishment of the Hardy-Weinberg principle (see below).

### 2.1 Basic biology: Genes and proteins

According to the central dogma of molecular biology, biological information is stored in the form of a sequence of nucleotides within a DNA molecule. This information is used (via transcription into mRNA followed by translation) to control the amino acid sequence of protein molecules and hence their 3d structure and function. It is therefore proteins that determine the characteristics of a living organism, while the information in the DNA sequence encodes the nature of the proteins that the organism can make. Although we now know that the true picture is significantly more complex [10], this concept suffices for the purpose of these notes.

The entire DNA sequence of an organism is known as its **genome**. Within the genome, a sequence of nucleotides that is transcribed into a functional RNA is known as a **gene** and the location of a particular gene along the entire sequence of DNA of the organism is known as a **locus**. Here we focus on genes that are transcribed into mRNA that is translated into protein – although we note that other types of genes also exist, encoding other types of functional RNA. Since protein molecules generally have specific functions, a protein-coding gene can often be associated with particular characteristics, such as immune system function, skin pigmentation or eye color. Within a gene, not all of the nucleotides actually encode protein sequence. Coding regions of the gene, also called exons, are transcribed into mRNA and translated into protein, while non-coding regions, or introns, are not transcribed, but are thought to fulfil functions such as regulation of transcription.

## 2.2 Alleles

The sequence of amino acids that makes up a given protein molecule is not unique, but can vary among individuals in a population. For example, one amino acid within the protein may be exchanged for another, or whole chunks of the amino acid sequence may be replaced, added or removed. Different protein variants may have different functional performance, leading to different characteristics of the organism. This protein-level variation arises (mostly) from differences among individuals in the DNA sequence of the relevant gene. Different sequence variants of the same gene are known as **alleles**. When population geneticists refer to "different alleles at the same locus" they mean alternative variants of the same gene (and hence alternative variants of the protein molecule encoded by the gene). However, as noted above, population genetic theory was developed prior to this molecular-level understanding: so the concepts of "allele" and "locus" were conceived, and can still be used, in a more abstract and generic way.

## 2.3 Genotype and phenotype

An organism's **genotype** is the list of alleles at all the loci within its genome. For example, the genotype of a human might include alleles encoding both brown eyes and blue eyes (inherited from the mother and father). The genotype is a description of the information that has been inherited by the organism; it is not a description the organism's actual characteristics.

The observable characteristics, or traits, of an organism are known as its **phenotype** ("pheno" comes from the Greek word meaning "observe"). The word phenotype can refer to anything from height or eye color, to the presence or absence of a disease. An organism's phenotype is often **heritable**, in other words, it is linked to genotype (as we discuss below). However, phenotype can also be influenced by other factors, from molecular effects (e.g. epigenetic modifications such as DNA methylation) to environmental and lifestyle factors, as well as pure chance. Flamingos are a classic example: their pink colouring is not inherited but is instead caused by pigments in their diet. A second example is human skin color. Our genes control the amount and type of melanin pigment in our skin, but our skin colour is also affected by exposure to UV light, which causes melanin to darken.

## 2.4 Link between genotype and phenotype

Humans have approximately 20,000 protein-coding genes. Most of them are present in duplicate copies (one from the father, one from the mother).[2] Therefore humans are **diploid** organisms, meaning that they have 2 alleles at each locus. Organisms that carry only one copy of their DNA (i.e. have only one allele at each locus) are known as **haploid**; this is the case for many bacteria. Some animals and plants carry 4 or even more copies of their genome; this is known as polyploidy.

Focusing on diploid organisms, the two alleles at a given locus, inherited from the mother and father, may be the same or different. If an individual inherits the same allele from their mother and father, their genotype is said to be **homozygous** at that locus. In contrast if the individual inherits two different alleles, the genotype is **heterozygous** for that locus.

An individual's genotype influences their phenotype (although, as explained above, other factors also play a role). However, the phenotype is not simply a sum of the effects of all the alleles carried by the individual: this is because the presence of one allele may suppress

---

[2]Human DNA consists of 46 separate (very long) strands, known as chromosomes. The 46 chromosomes can be grouped into 23 pairs, of which 22 are duplicate pairs, one copy being inherited from the mother and one from the father. The last chromosome pair, the sex chromosomes, also consists of 2 duplicate copies in females, but in males, a short "Y" chromosome containing only 107 protein-coding genes is inherited from the father. Therefore, in males, genes on the sex chromosomes are not present in duplicate copies.

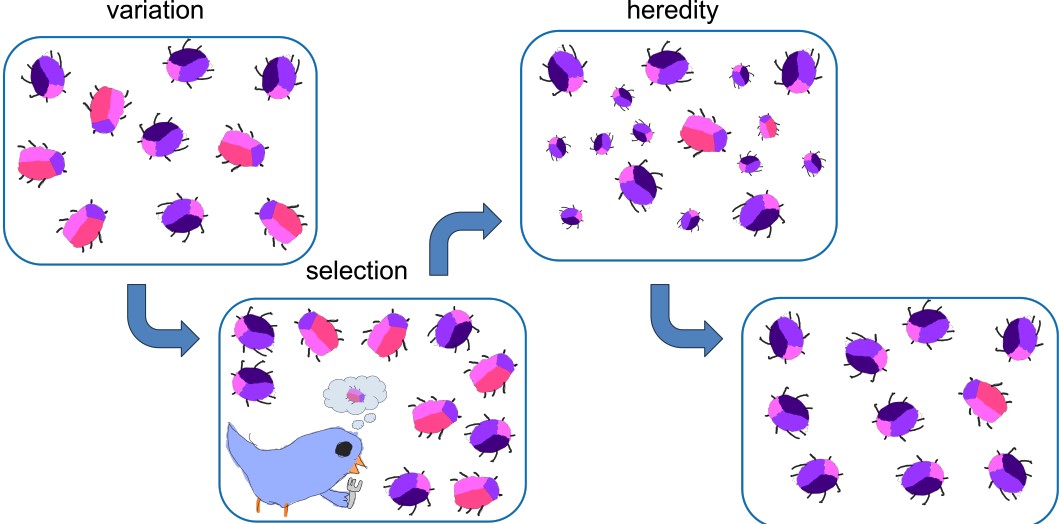

Figure 1: A cartoon illustrating the theory of natural selection (graphics: Naomi Verhoek).

the phenotypic effects of another. For example, a person might carry one allele for brown eye colour and another for blue eye colour (they are heterozygous). This individual does not (generally) have one blue and one brown eye but instead both eyes are brown. This is because the allele for brown eye colour is **dominant** while the allele for blue eye colour is **recessive**. On the other hand, if the individual is homozygous, with two copies of the allele for blue eye colour, they will have blue eyes. This is known as Mendelian inheritance and it will be discussed in more detail later in these notes. From a molecular point of view, this phenomenon arises from the fact that not all of the genes encoded in an individual's DNA are expressed (i.e. transcribed and translated into protein molecules), as well as the fact that protein molecules can interact with each other. The individual's phenotype is influenced only by those genes that are actually expressed, as well as by the interactions between the various protein molecules that the organism produces.

To make things more complicated, many phenotypes are actually **polygenic**; they are not determined by the alleles that are present at a single locus, but instead depend on multiple loci. Taking a more in depth look at human eye colour, in fact this is determined by approximately 16 different genes – although 2 of these genes play the major role.

## 3 The theory of natural selection

Population genetic theory predicts mathematically how populations change in time under the influence of natural selection. Therefore we start our discussion with a brief introduction to natural selection.

The central tenets of the theory of natural selection are that (i) individuals within a population vary in phenotype, (ii) phenotype is heritable, and (iii) phenotype is coupled to differential reproduction, i.e. individuals have more or fewer offspring, on average, depending on their phenotype.

Figure 1 illustrates in cartoon form how these principles can cause the characteristics of a species to change over time. A beetle population shows phenotypic variation in coloration: some beetles are pink/red while others are purple/black (note these are not different species, but rather phenotypic variants within the same species). A predatory bird species prefers

to feed on pink/red beetles, and is less likely to feed on purple/black beetles. Due to the differential predation pressure, purple/black beetles have, on average more offspring, and since colouration is hereditary, the offspring of purple/black beetles tend to be purple/black. Therefore the population shifts over time towards a greater fraction of purple/black beetles, i.e. the colouration phenotype of the beetle species gradually changes, driven by predation pressure from the birds.

> **Summary of chapter 3**
>
> - The theory of natural selection states that species change in time because heritable phenotypes vary among individuals in the population, and are linked to differential reproductive success.
>
> - Population genetic theory predicts mathematically how natural selection acts on populations.

## 4 The maintenance of variation in populations

One of the reasons why the theory of natural selection was not initially widely accepted was that it requires that there is phenotypic variation among individuals within a species, but it does not explain where the phenotypic variation comes from. To understand why populations show phenotypic variation that natural selection can act on, we have to discuss in more detail the rules that govern heredity of phenotypes for diploid organisms.

### 4.1 Blending inheritance

At the time of Darwin, the molecular aspects of inheritance that we discussed in section 2 were completely unknown. The prevailing idea was that of **blending inheritance**, in which progeny exhibit the average phenotype of their two parents. Fleeming Jenkin (1833-1885), a critic of Darwin, pointed out that blending inheritance is not consistent with the existence of phenotypic variation in populations, and hence is not consistent with the theory of natural selection [11].

To see this, let us suppose that a population of individuals has some phenotype $x$ that can be quantified (such as height of the individuals). At time zero, the phenotype varies among individuals such that its probability distribution is $f(X)$ with mean $\mu$ and variance $\sigma^2$. Now we suppose that random pairs of individuals within the population mate and produce offspring. The phenotype values $x_1$ and $x_2$ of a given set of parents are independently sampled from the probability distribution $f(X)$. Under blending inheritance, the phenotype value $x_{\text{new}}$ of the offspring will be a random variable that is the average of $x_1$ and $x_2$:

$$X_{\text{new}} = \frac{X_1 + X_2}{2}.$$

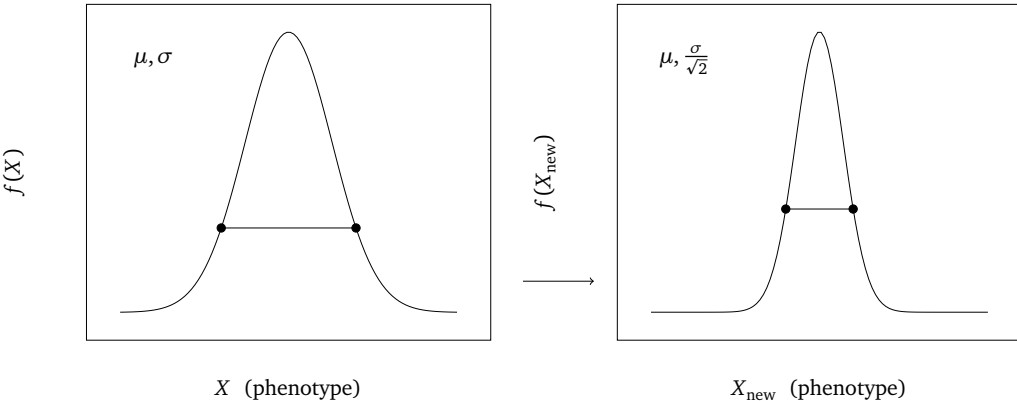

Figure 2: Schematic illustration of the probability distribution $f(X)$ of the phenotype value $x$ before and after one generation of blending inheritance, under the assumption of random mating. The mean phenotype value $\mu$ in the offspring generation is the same as that of the parent generation but the standard deviation $\sigma$ of the phenotype value is smaller by a factor of $\sqrt{2}$ in the offspring generation. A Gaussian form of $f(X)$ is sketched here to illustrate the changes in $\mu$ and $\sigma$, but the distribution $f(X)$ can in principle take any form.

Using basic probability theory [12], the mean and variance of the phenotype value of the offspring are given by

$$\mu_{\text{new}} = \frac{\cancel{2}\mu}{\cancel{2}} = \mu, \qquad \sigma_{\text{new}}^2 = \frac{\cancel{2}\sigma^2}{\cancel{4}2} = \frac{\sigma^2}{2}.$$

Figure 2 illustrates this result. Each generation of blending inheritance reduces the standard deviation $\sigma$ of the phenotype value across the population by a factor of $\sqrt{2}$. It is easy to see that after many generations $\sigma$ will become so small that all individuals in the population will essentially have the same phenotype value (i.e. $f(X)$ will tend to a delta function at $\mu$).

Therefore blending inheritance rapidly removes phenotypic variation, leading to a phenotypically uniform population. If a population is phenotypically uniform then natural selection cannot change the average phenotype over time, since all individuals produce the same number of offspring on average.

## 4.2 Mendelian inheritance

Unbeknownst to Darwin, the answer to this puzzle was being discovered virtually contemporaneously by the Austrian-Czech scientist and monk Gregor Mendel (1822-1884). Mendel published his findings in 1865 [13], only 7 years after Darwin's famous book *The Origin of Species* [7], but unfortunately his work was largely ignored during his lifetime. Its relevance for the theory of natural selection was realised decades later; indeed, Mendel is now widely regarded as the founder of modern genetics.

Mendel performed extensive breeding experiments with peas in his monastery garden, and he documented the proportions of offspring with particular phenotypes in different generations. His key finding was that, for some phenotypes, blending inheritance does not hold. For example, if a pure-bred pea plant with red flowers is mated with a pure-bred pea plant with white flowers, the offspring do not produce pink flowers. Instead, all the offspring in the first generation (known as the "F1 generation") produce red flowers. However, if these offspring are mated with each other, then 75% of the plants in the next ("F2") generation will have red flowers, while the remaining 25% of the F2 plants will have white flowers (even though both of their F1 parents had red flowers).

To rationalize his data, Mendel suggested that each individual inherits two "factors" (later known as alleles), one from the mother and one from the father, and that these can be dominant or recessive. If an individual inherits two identical alleles (i.e. it is homozygous; see section 2) then it will have the corresponding phenotype. However if it inherits two different alleles (i.e. it is heterozygous), it will show the phenotype of the dominant allele.

To show how this hypothesis is consistent with Mendel's data on the ratios of red and white flowered pea plants in the F2 generation, we consider the mating of two parents that are heterozygous at some locus, with possible alleles $A_1$ and $A_2$, in other words, both parents have genotype $A_1 A_2$. Assuming that the alleles of the parents are randomly distributed to the offspring, we can construct a table, called a Punnett Square,[3] to show the possible offspring genotypes:

|       | $A_1$     | $A_2$     |
|-------|-----------|-----------|
| $A_1$ | $A_1 A_1$ | $A_1 A_2$ |
| $A_2$ | $A_1 A_2$ | $A_2 A_2$ |

In the Punnett Square, the parental genotypes are shown along the top and to the left of the table, while the 4 entries in the table show the outcomes of each possible combination of parental alleles. We see that the offspring of this mating have genotypes $A_1 A_1$, $A_1 A_2$ or $A_2 A_2$ with probabilities 0.25 : 0.5 : 0.25, consistent with Mendel's observations.

Mendel's hypothesis has the important feature that alleles do not mix with each other – rather, they remain in pure form through the generations. Furthermore, their associated phenotype may reappear in later generations even after it has apparently disappeared from the population. At the time, Mendel did not know what alleles actually were physically. This knowledge was only gained approximately 100 years later, after the discovery of DNA as the unit of inheritance.

## 4.3 Polygenic phenotypes

While some phenotypes do follow Mendelian inheritance (a famous example being the incidence of the disease sickle cell anemia in humans [10]), others apparently do not. For example, human height, weight and hair colour all show continuous variation, such that offspring can show phenotypes that are a "mixture" of those of their parents. These are polygenic phenotypes, which are controlled by multiple genetic loci. The "infinitesimal model", proposed in 1918 by Ronald Fisher (1890-1962) shows how the existence of continuously varying traits is consistent with Mendelian inheritance [14]. The details are beyond the scope of these notes (although Ref. [15] provides a detailed treatment). For our purpose we merely comment that the infinitesimal model demonstrates that, if many loci contribute to a phenotype, the random sampling of alleles at each locus produces a continuous, normal distribution of phenotype values in the population (this is essentially an example of the central limit theorem).

## 4.4 The Hardy-Weinberg principle

We now return to the central question of this section: how is phenotypic variation maintained in populations, so that natural selection can act on it? We already saw that blending inheritance does not maintain phenotypic variation, and is therefore inconsistent with natural selection. We now demonstrate that Mendelian inheritance does maintain phenotypic variation, and hence it is consistent with natural selection. This fact was established in 1908 and underlies the theory of population genetics. It is known as the **Hardy-Weinberg Principle**, after its discoverers, the British pure mathematician G. H. Hardy (1877-1947) and the German obstetrician Wilhelm Weinberg (1862-1937).

---

[3]The Punnett Square was invented in 1905 by the geneticist Reginald C Punnett (1875-1967).

We start by assuming a large population of diploid individuals, who choose mating partners at random. To keep things simple, we will assume that individuals are monoecious (any individual can be interchangeably male or female[4]). We suppose that at a particular genetic locus, 2 alleles are possible, denoted $A_1$ and $A_2$. Therefore individuals have 3 possible genotypes: $A_1A_1$, $A_1A_2$ and $A_2A_2$. The starting proportions of these genotypes in the population are denoted $P$, $2Q$ and $R$ (the factor of 2 arises because $A_1A_2$ is equivalent to $A_2A_1$). While any set of starting genotype proportions is possible, clearly we must have $P + 2Q + R = 1$.

We aim to calculate how the genotype proportions change over time as organisms mate with each other and produce offspring. We denote the proportions after one round of mating with a single prime, and the proportions after two rounds of mating with a double prime.

|  | $A_1A_1$ | $A_1A_2$ | $A_2A_2$ |
|---|---|---|---|
| at start | P | 2Q | R |
| first mating | P' | 2Q' | R' |
| second mating | P'' | 2Q'' | R'' |

We start by considering the first round of mating. From the proportions ($P$, $2Q$, $R$) of the 3 genotypes that are present in the population, we can calculate the probability of mating between each possible genotype pair (shown in black in the table):

| Parent 2 | | Parent 1 $A_1A_1$ $P$ | $A_1A_2$ $2Q$ | $A_2A_2$ $R$ |
|---|---|---|---|---|
| $A_1A_1$ | $P$ | $P^2$ | $2PQ$ | $PR$ |
| $A_1A_2$ | $2Q$ | $2PQ$ | $4Q^2$ | $2QR$ |
| $A_2A_2$ | $R$ | $PR$ | $2QR$ | $R^2$ |

Next, for every possible mating, we predict the proportions of offspring genotypes under Mendelian inheritance using a Punnett square:

$A_1A_1$ x $A_1A_1$

|  | $A_1$ | $A_1$ |
|---|---|---|
| $A_1$ | $A_1A_1$ | $A_1A_1$ |
| $A_1$ | $A_1A_1$ | $A_1A_1$ |

$A_1A_2$ x $A_1A_1$

|  | $A_1$ | $A_2$ |
|---|---|---|
| $A_1$ | $A_1A_1$ | $A_1A_2$ |
| $A_1$ | $A_1A_1$ | $A_1A_2$ |

$A_2A_2$ x $A_2A_2$

|  | $A_2$ | $A_2$ |
|---|---|---|
| $A_2$ | $A_2A_2$ | $A_2A_2$ |
| $A_2$ | $A_2A_2$ | $A_2A_2$ |

$A_1A_1$ x $A_2A_2$

|  | $A_1$ | $A_1$ |
|---|---|---|
| $A_2$ | $A_1A_2$ | $A_1A_2$ |
| $A_2$ | $A_1A_2$ | $A_1A_2$ |

$A_1A_2$ x $A_2A_2$

|  | $A_1$ | $A_2$ |
|---|---|---|
| $A_2$ | $A_1A_2$ | $A_2A_2$ |
| $A_2$ | $A_1A_2$ | $A_2A_2$ |

$A_1A_2$ x $A_1A_2$

|  | $A_1$ | $A_2$ |
|---|---|---|
| $A_1$ | $A_1A_1$ | $A_1A_2$ |
| $A_2$ | $A_1A_2$ | $A_2A_2$ |

The probability $P(g)$ of observing a particular genotype $g$ in the next generation is given by

$$P(g) = \sum_m P(g|m) \cdot P(m),$$

where the index $m$ runs over the possible matings (i.e. combinations of parental genotypes), $P(m)$ is the probability of a given mating and $P(g|m)$ is the probability of an offpring of genotype $g$ being produced from mating $m$.

---

[4]Cucumber plants are an example of a monoecious organism.

Using the results for $P(m)$ and $P(g|m)$ computed in the table and Punnett squares above, we obtain the following genotype proportions after one round of mating:

$$
\begin{aligned}
P' &= P(A_1 A_1) = (P + Q)^2, \\
2Q' &= P(A_1 A_2) = 2(P + Q)(Q + R), \\
R' &= P(A_2 A_2) = (Q + R)^2.
\end{aligned} \tag{1}
$$

We can now use this result to compute the genotype proportions $P'', 2Q'', R''$ after a second round of mating. To do this we simply substitute $P'$ for $P$, $Q'$ for $Q$ and $R'$ for $R$ into Eqs (1):

$$
\begin{aligned}
P'' &= (P' + Q')^2, \\
2Q'' &= 2(P' + Q')(Q' + R'), \\
R'' &= (Q' + R')^2.
\end{aligned} \tag{2}
$$

Next, we substitute Eqs (1) into Eqs (2), and also make use of $P + 2Q + R = 1$ to obtain the key result:

$$
\begin{aligned}
P'' &= P', \\
Q'' &= Q', \\
R'' &= R'.
\end{aligned} \tag{3}
$$

Therefore, after one round of mating, the genotype proportions reach a steady state (the "Hardy-Weinberg equilibrium") which remains unchanged by subsequent rounds of mating. Therefore this calculation shows that genetic diversity is maintained in a population under Mendelian inheritance (this is the Hardy-Weinberg principle).

It is also useful to consider the relative abundances of different alleles in the population. Considering allele $A_1$, genotype $A_1 A_1$ (with frequency $P$) has 2 copies of this allele, while genotype $A_1 A_2$ (with frequency $2Q$) has 1 copy. The total number of $A_1$ alleles in the population is therefore $2N(P + Q)$, where $N$ is the number of individuals. Likewise the total number of $A_2$ alleles in the population is $2N(Q+R)$. Since the total number of alleles in the population is $2N$, the fractional abundance of allele $A_1$ is $P + Q$ and the fractional abundance of allele $A_2$ is $Q+R$. It is traditional to define these fractional allele abundances as $p$ and $q$, where $p \equiv P+Q$ and $q \equiv Q+R$. Naturally, $p + q = 1$.

An interesting result emerges when we express the steady-state genotype abundances $P''$, $2Q''$ and $R''$ in terms of the fractional allele abundances $p$ and $q$:

| genotype | $A_1 A_1$ | $A_1 A_2$ | $A_2 A_2$ |
|---|---|---|---|
| steady state abundance | $p^2$ | $2pq$ | $q^2$ |

This result shows that the Hardy-Weinberg equilibrium (i.e. the steady state genotype abundances that are reached after multiple rounds of mating under Mendelian inheritance) is simply equivalent to random assortment of alleles among individuals in the population.

**Summary of chapter 4**

- Blending inheritance, in which the offspring takes the average phenotype of its parents, rapidly eliminates phenotypic variation from the population.

- In Mendelian inheritance, offspring inherit two alleles at each locus, one from each parent; the phenotype of a heterozygote is that of the dominant allele.

- The Hardy-Weinberg principle shows that variation is maintained in a population under Mendelian inheritance.

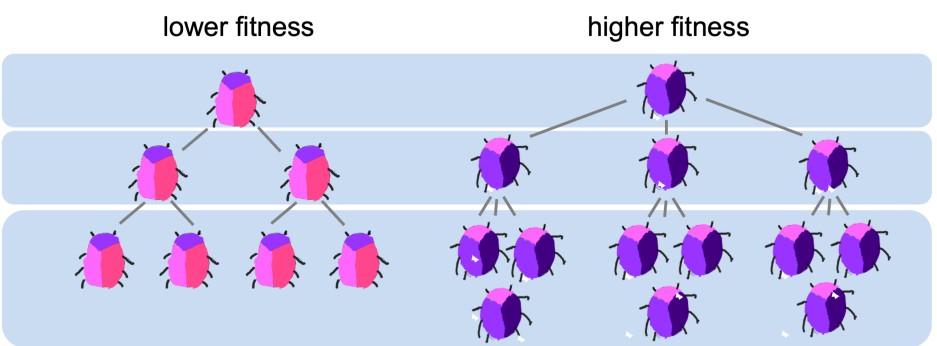

Figure 3: Illustration of the concept of fitness (graphics: Naomi Verhoek). Purple-black beetles produce more offspring per individual than pink-red beetles, therefore purple-black beetles have higher fitness and pink-red beetles have lower fitness. In this example, the ratio of abundances $R = N_{\text{pink−red}}/N_{\text{purple−black}}$ is 1 at the start, 2/3 after one generation and 4/9 after two generations. So the relative fitness $w_{\text{pink−red}}$ of the pink-red beetles is 2/3 and the selection coefficient $s_{\text{pink−red}}$ is -1/3. Conversely the relative fitness of the purple-black beetles $w_{\text{purple−black}}$ is 3/2 and the selection coefficient $s_{\text{purple−black}}$ is 1/2.

## 5 Mutation and selection

In chapter 4 we discussed two of the components of the theory of natural selection – maintenance of phenotypic variation within a population, and heredity of phenotypes. To understand fully how populations change under natural selection, we need to consider the final component of the theory – differential reproduction of different phenotypes (selection; Figure 1), and we also need to discuss the ultimate source of variation within the population (mutations).

### 5.1 The concept of fitness

Natural selection aims to explain how evolution causes species to become optimized in their environment. In population genetics, "optimization" is measured by the concept of **fitness**. Fitness quantifies the relative reproductive success of an organism, compared to other organisms. It is defined as the number of offspring that organisms of a particular genotype/phenotype leave behind, on average, relative to organisms of another genotype/phenotype.

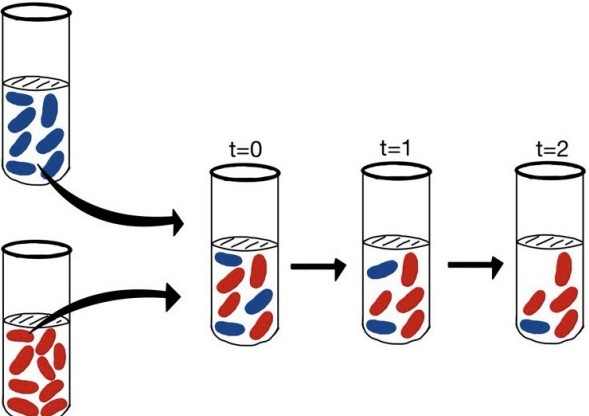

Figure 4: Illustration of a competition experiment. Two bacterial strains (with different genotypes) are mixed at time $t = 0$ in nutrient medium and are allowed to grow together over several generations. The abundance of bacteria of each strain is measured at the start and at regular time intervals. If the two strains differ in fitness, the ratio of their abundances is expected to change in time.

Specifically, let us imagine a mixed population containing organisms with two different genotypes A and B. The numbers of organisms of genotypes A and B are $N_A$ and $N_B$ respectively, and the ratio of the abundances of the two genotypes is $R = N_B/N_A$. We now track how $N_A$, $N_B$ and the ratio $R$ change in successive generations. If organisms of type A and B have different reproductive success, then in each generation $N_A$ and $N_B$ will change by different factors, so the ratio $R$ will change. The relative fitness is defined as the factor by which $R$ changes per generation.

For example, let us suppose that for every individual of genotype A there are, on average, 3 individuals of that genotype in the next generation, but for every individual of genotype B there are only 2 type B individuals in the next generation (as illustrated in Figure 3). If we start with equal numbers of individuals of types A and B, the initial ratio $R(0) = 1$. In the next generation, the ratio will become $R(1) = 2/3$. Therefore the relative fitness of genotype B is $w_B = R(1)/R(0) = 2/3$, while the relative fitness of genotype A is $w_A = 3/2$.

## 5.2 The selection coefficient

It is often convenient to use a slightly different measure of relative fitness: the **selection coefficient (s)**. This is just defined as $s = w - 1$. The sign of $s$ indicates whether the organism of interest is fitter or less fit than its competitor. In the example above, if genotype B is fitter than A, we have $w_B > 1$ and hence $s_B > 0$, whereas if B is less fit than A, we have $w_B < 1$ hence $s_B < 0$.

The selection coefficient can be measured in a **competition experiment**. Figure 4 illustrates a competition experiment between two bacterial strains.[5] Let us suppose that these are the wild-type strain (WT) and a mutant (M) whose genotype differs from the wild-type, e.g. because a particular gene has been altered. Denoting the (time-dependent) number of wild-type and mutant bacteria as $N_{WT}(t)$ and $N_M(t)$, the ratio of mutant/wild type is $R(t) = N_M(t)/N_{WT}(t)$. The selection coefficient $s$ can be extracted from the rate of change of

---

[5]Competition experiments are easier to perform for micro-organisms such as bacteria than for animals or plants. Bacteria also evolve faster than animals or plants, making them attractive models for testing evolutionary theory. Furthermore, understanding how bacterial populations evolve is clinically relevant, given the threat of antimicrobial resistance.

$R(t)$. To see this, we first consider how $R$ changes over discrete generations. Using the definitions of the fitness $w$ and selection coefficient $s$ of the mutant strain relative to the wild-type strain, we can write that after 1 generation $R(1) = wR(0) = R(0)(1+s)$, after 2 generations $R(2) = wR(1) = R(0)(1+s)^2$ and after $\tau$ generations

$$R(\tau) = wR(\tau - 1) = R(0)(1+s)^\tau .$$

Taking logs we obtain $\log R(\tau) = \log R(0) + \tau \log(1+s)$, and using the fact that for small $s$, $\log(1+s) \simeq s$, we obtain

$$s \simeq \frac{\log R(\tau) - \log R(0)}{\tau},$$

and hence

$$s = \frac{\Delta \log R}{\Delta \tau}.$$

The selection coefficient is therefore the rate of change of the logarithm of the ratio $R$, for time measured in generations.

## 5.3 The (lack of) speed of selection

Now that we have properly defined the Darwinian concept of differential reproduction, i.e. selection, we can ask how selection changes the proportions of different genotypes within a diploid population under Mendelian inheritance. This question was first addressed by the British-Indian polymath **J.B.S. Haldane** (1892-1964).

We repeat the analysis of section 4.4, this time including selection. We consider again a diploid, randomly mating, monoecious population of size $N$ individuals, with 2 alleles $A_1$ and $A_2$ at the genetic locus of interest. The population is taken to be initially at Hardy-Weinberg equilibrium, i.e. the starting proportions of the 3 possible genotypes $A_1A_1$, $A_1A_2$ and $A_2A_2$ are

| genotype | $A_1A_1$ | $A_1A_2$ | $A_2A_2$ |
|---|---|---|---|
| relative frequency | $p^2$ | $2pq$ | $q^2$ |

where $p$ and $q$ are the fractional abundances of alleles $A_1$ and $A_2$.

To include the effects of selection, we suppose that $A_1$ is a recessive mutant allele. The homozygote $A_1A_1$ has fitness $w = 1+s$, whereas the fitness of the other genotypes $A_1A_2$ and $A_2A_2$ is 1. We will suppose that the fitness difference arises because offspring of genotype $A_1A_1$ have differential survival probability compared to offspring of the other genotypes.

After one round of mating, the number of individuals of each genotype in the population will be:

| genotype | $A_1A_1$ | $A_1A_2$ | $A_2A_2$ |
|---|---|---|---|
| number of individuals | $Np^2(1+s)$ | $2Npq$ | $Nq^2$ |

This result can be derived by repeating the analysis of section 4.4, accounting for the fact that the probability of offspring of genotype $A_1A_1$ surviving is different to that of the other genotypes.[6]

It is important to note that the total population size has changed during this round of mating: it is now $N[p^2(1+s) + 2pq + q^2] = N(1+sp^2)$ (where we have used $p+q=1$).

---

[6]One can also derive the same result by noting that the Hardy-Weinberg equilibrium genotype proportions would be maintained in the absence of selection; the effect of selection is just to scale the number of individuals of genotype $A_1A_1$ by a factor $(1+s)$.

To find the relative frequencies of the genotypes after one round of mating, we need to scale by the population size:

| genotype | $A_1A_1$ | $A_1A_2$ | $A_2A_2$ |
|---|---|---|---|
| relative frequency | $p^2(1+s)/(1+sp^2)$ | $2pq/(1+sp^2)$ | $q^2/(1+sp^2)$ |

The effect of selection is to shift the genotype frequencies away from the Hardy-Weinberg equilibrium. The genotype frequencies are no longer in steady state but instead change from generation to generation.

To understand better the speed at which selection shifts the genotypic composition of the population, let us calculate the relative abundance of the allele $A_1$ in the population. For a population without selection, this is $p$, as calculated in section 4.4. After one generation with selection, it is given by

$$p_{\text{new}} = \frac{p^2(1+s)}{1+sp^2} + \frac{pq}{1+sp^2} = \frac{p(ps+1)}{1+sp^2}.$$

The rate of change of the allele frequency per generation is then given by

$$\Delta p = p_{\text{new}} - p = \frac{sp^2q}{1+sp^2}. \tag{4}$$

If the selection coefficient is positive ($s > 0$), indicating that allele $A_1$ increases the fitness of the homozygote, we see that $\Delta p > 0$, i.e. the frequency of the allele increase over time in the population. Conversely, if $s < 0$, indicating that allele $A_1$ is detrimental, $\Delta p < 0$, and the frequency of the allele decreases over time.

Typically, selection coefficients are small, $s \simeq 0.01$. If we suppose that a mutant allele with a selection coefficient $s = 0.01$ is present at a frequency of about 1% in the population (a reasonable scenario), then we have $p = 0.01$ and $q = 0.99$. Substituting these values into Eq. (4), we find that $\Delta p \simeq 10^{-6}$ per generation. In other words, hundreds to thousands of generations of selection are required to significantly change the abundance of the mutant allele within the population. Therefore we arrive at the important conclusion that selection, though it does change genotype abundances over time, typically acts very slowly.

## 5.4 Mutations and how they happen

Selection, as discussed in sections 5.2 and and 5.3, increases the proportion of fitter genotypes in the population, but it does not create new variation. What is the source of genetic variation?

Variation within a population is ultimately created by **mutations**: changes in the sequence of DNA. Since the DNA sequence encodes the amino acid sequence of proteins that ultimately determine phenotype, mutations can (occasionally) lead to new phenotypes.

Mutations can take different forms. Here we focus on changes to single base pairs within the DNA genome ("point mutations"), but mutations can also involve deletion or insertion of different-sized "chunks" of DNA sequence.

Figure 5 shows an example of how a single base pair change can lead to a new phenotype. A point mutation from A to T in the human gene encoding the protein molecule haemoglobin causes a glutamate amino acid (whose codon is GAG) to be replaced by a different amino acid, valine (whose codon is GTG). Haemoglobin is found in red blood cells where it binds to oxygen, allowing it to be transported in the blood from the lungs to the muscles. The substitution of glutamate by valine at this particular position in the protein causes haemoglobin molecules to clump together, affecting the shape and function of the red blood cells. People who have this mutation on both copies of their haemoglobin gene experience sickle cell anaemia.

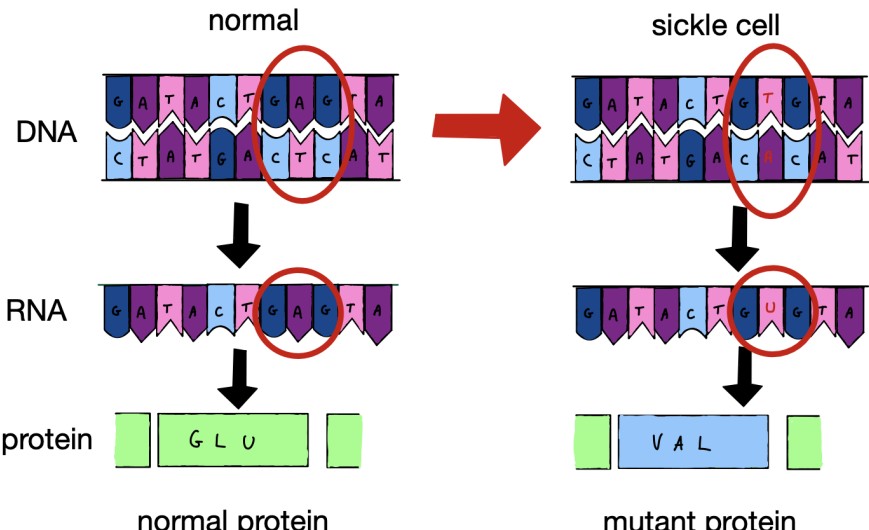

Figure 5: A point mutation is the cause of sickle-cell anaemia. The illustration shows a part of the human gene encoding the protein haemoglobin. A mutation that changes one base-pair in this sequence (changing a GAG codon, encoding glutamate, to a GTG, encoding valine) causes a glutamate in the haemoglobin molecule to be replaced by a valine. If a person has this mutation on both copies of their haemoglobin gene, they will have sickle cell anaemia.

Once a mutation has happened, it can be inherited by offspring. Indeed, the inheritance patterns of the sickle cell anaemia phenotype within families are a classic example of Mendelian inheritance.

From a molecular point of view, point mutations often happen due to mistakes made by the molecular machinery that copies the DNA genome when cells proliferate. Cells have mechanisms to correct such errors, but these occasionally fail, leading to mutations. DNA replication is not the only source of point mutations: they can also be caused by damage to the DNA due to UV light or some chemicals. When such damage happens, the cell's DNA repair machinery sometimes makes an incorrect repair, leading to an altered DNA sequence. Since cancer is caused by mutations, this explains why exposure to UV light and/or various chemicals can be a risk factor for cancer.

## 5.5 The rate of mutations

To make quantitative models for evolution, it is important to measure how often mutations happen. However, this turns out to be harder than it sounds. Firstly, the mutation rate depends on many factors, including the type of mutation and its position in the genome, the physiological state of the organism (e.g. whether it is stressed), chemical or other factors in the environment, and other mutations the organism might have (e.g. mutations in the DNA repair system can greatly increase an organism's mutation rate). Secondly, measuring the rate of mutations is, from a practical point of view, not easy.

The classical method for measuring mutation rates was established for bacteria in 1943 by the microbiologist Salvador Luria (1912-1991) and the biophysicist Max Delbrück (1906-1981). It is known as the **Luria-Delbrück experiment** (or "fluctuation test") [16]. The Luria-Delbrück experiment not only provided a way to measure mutation rates, but also proved

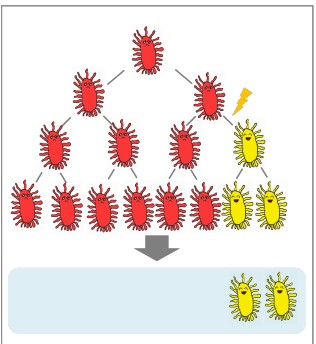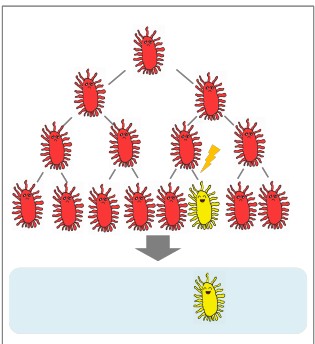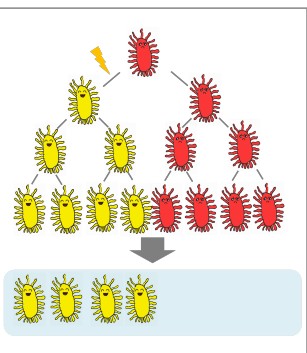

Figure 6: Schematic illustration of the Luria-Delbrück experiment (graphics: Naomi Verhoek). The lineage trees of 3 replicate exponentially-growing bacterial populations are shown. In each population, a mutation happens (orange jagged arrow). The mutation changes a red bacterium into a yellow bacterium, and all descendants of the mutated yellow bacterium are also yellow. Red bacteria are antibiotic-sensitive while yellow bacteria are resistant to antibiotic. After 3 generations, the population is exposed to antibiotic (blue shading), so that only yellow bacteria survive and are counted. The count of yellow bacteria varies widely among the replicate experiments, depending on when the mutation happened in the lineage tree. Although not shown here, some replicate experiments may not produce any resistant bacteria.

definitively that mutations can arise in growing populations in the absence of any selection. In this experiment, illustrated in Figure 6, multiple replicate populations are allowed to grow for *N* generations. During this growth, mutations accumulate. The number of mutants in each replicate population is then measured by phenotypic testing. For example, if one were interested in mutations that lead to resistance to a particular antibiotic, one would spread each replicate bacterial population onto an agar plate containing a high dose of antibiotic and count how many bacteria are able to grow into colonies (implying that they are resistant to the antibiotic; Figure 6). The statistical distribution of the number of mutants, measured across the replicate populations, is then fit to a mathematical model to estimate the mutation rate. However, this is a notoriously imprecise procedure. First, it relies on the mathematical model being correct, which may not be the case. Second, the mutant number distribution itself is hard to measure accurately because it is long-tailed. In other words, most replicate populations only have a few mutants but some of the replicates have a very large number. These are known as "jackpot events". They happen when a mutation occurs at an early stage in the growth experiment: because the population is growing exponentially, these mutants will multiply exponentially and accumulate many descendants by the end of the experiment. In contrast, mutations that happen late in the experiment only accumulate a few descendants. Interesting recent work has focused on how the mutant number distribution is altered in different cases, including spatially structured populations [17] and mutations that show phenotypic effects only after a time delay [18].

Technological advances over the past decades have greatly increased the speed at which DNA can be sequenced, and greatly decreased the price of such endeavours. This has made possible a more direct approach to measuring mutation rates, in which one simply sequences the whole genome of the organism periodically, as the population grows and accumulates mutations. For the bacterium *Escherichia coli*, comparison of this approach with the Luria-Delbrück methods suggests that the true mutation rate may be about a factor of 10 higher than that estimated in the Luria-Delbrück experiment [19].

While bearing in mind all these caveats, it is still useful to provide a ballpark number. For humans, a typical mutation rate is $10^{-6}$ per gene per generation. If we assume a very simple model in which a population contains $N$ that genes switch from normal to mutated form at a rate $\mu = 10^{-6}$ per generation, the fraction $f$ of mutant genes obeys

$$\frac{df}{dt} = \mu(1-f),$$

which can be solved to show that the timescale over which the population's genotype composition changes due to mutations is $1/\mu \simeq 10^6$ generations. Therefore mutations change populations over time, but only very slowly.

## 5.6 The fate of mutations in the population

A new mutation typically arises in only a single individual. That individual may or may not pass the mutation on to the next generation, after which it may or may not spread within the population. Which of these outcomes happens depends on the fitness effect of the mutation and also (to a large extent) on pure chance.

The fitness effects of mutations can be categorized as (i) neutral, (ii) deleterious or (iii) beneficial. Neutral mutations have no detectable effect on fitness. This might be the case if the mutation happens in a non-coding part of the genome, or, if it happens in a coding region, it changes the DNA sequence without changing the encoded amino acid (a synonymous mutation). Deleterious mutations reduce the fitness of the organism. This might happen if the gene for a functional enzyme is changed in a way that makes it less effective (or even non-functional). In the extreme case, deleterious mutations can be lethal, implying that the mutated organism is incapable of life. Beneficial mutations increase the fitness of the organism, for example by changing a protein so that it functions better under the conditions where the organism finds itself, or by increasing the expression level of a beneficial protein.

The **distribution of fitness effects** is a function describing what proportion of mutations are advantageous, neutral or deleterious. This function plays an important role in evolutionary theory, but it is typically hard to measure, since it requires the laborious measurement of fitness effects (that can be small) for very many mutants. In some cases it has been measured, e.g. for the RNA virus vesicular stomatitis virus [20], where random mutations were introduced and the fitness of the mutants assessed relative to the wild-type virus. This study showed that many random mutations are lethal, but of those that are not lethal, most are neutral, with only a few having small beneficial or detrimental effects.

Let us first think about the fate of a deleterious mutation, i.e. a mutation that decreases the fitness of the organism. These mutations have a negative selection coefficient $s$. As we discussed in section 5.3, they will be slowly removed from the population by selection; their frequency is expected to decrease at rate $sp^2q/(1+sp^2)$ per generation (Eq. 4). But at the same time, new deleterious mutations spontaneously appear at rate $Nq\mu$ where $N$ is the population size, $\mu$ is the mutation rate per generation and $q = 1 - p$ is the un-mutated fraction of the population. At steady-state, we expect a balance between these two processes. This can be expressed as

$$\frac{sp^2q}{1+sp^2} + \mu q = 0.$$

Using the fact that generally $s$ is small and $p$ is small, such that $sp^2 << 1$, and $q \approx 1$, the expression above can be simplified to $sp^2 + \mu = 0$. This leads to a simple result for the steady-state fraction $p^*$ of mutants in the population:

$$p^* = \sqrt{-\frac{\mu}{s}}.$$

Using the typical values $\mu \simeq 10^{-6}$ and $s \simeq 0.01$, we arrive at $p^* \simeq 1\%$. Therefore we expect that deleterious mutations will be maintained within the population, at a low frequency that reflects the balance between their creation by mutation and their removal by selection. This is known as **mutation-selection balance**.

Next, let us consider the fate of a beneficial mutation. Section 5.3 tells us that a beneficial mutation should slowly increase in frequency in the population, due to selection. However, in fact beneficial mutations often arise and then rapidly disappear without becoming established in the population. This is because the fate of a beneficial mutation is initially determined by stochastic effects. When the mutation first arises it is typically present in only one individual, and its fate depends critically on whether that individual (and its immediate offspring) survives, and if so how many offspring it produces, all of which is strongly affected by random chance. The probability that a beneficial mutation makes it through this early, stochastic phase of its existence, and achieves high enough frequency in the population for selection to take effect, is called the **fixation probability**. We do not calculate the fixation probability in these notes, but we will discuss stochastic effects in some detail in chapter 6.

> **Summary of chapter 5**
>
> - The concept of fitness is used to quantify optimality of a population.
>
> - Fitness $w$ measures relative number of offspring per individual. The selection coefficient is $s = w - 1$. Positive (negative) values of $s$ indicate increased (decreased) fitness relative to competitors.
>
> - The selection coefficient can be measured in a competition experiment. Selection increases the proportion of fitter genotypes within a population over time, but the rate of selection is slow for realistic selection coefficients.
>
> - Mutations are changes in the DNA sequence. They can have neutral, deleterious or beneficial effects on fitness. Mutations change the genotypic composition of a population only slowly.
>
> - Deleterious mutations are maintained in the population at a low level via mutation-selection balance. The fate of beneficial mutations is initially stochastic and determined by the fixation probability.

# 6 Stochastic effects: Genetic drift

In chapter 5 we noted that the fate of a new mutation is, initially, determined by stochastic birth and death events, since the mutation is present in only a small number of individuals. More broadly, stochastic effects play an important role in evolution in any situation where small numbers of individuals are involved. This could be the founding of a new population by a small number of randomly selected "pioneers" from a different habitat (known as founder effects), random birth and death events, randomness in offspring numbers in small populations, or stochastic environmental events that reduce the population size ("catastrophes"). In population genetic theory, all sources of stochastic fluctuations are grouped together and collectively referred to as **genetic drift**.

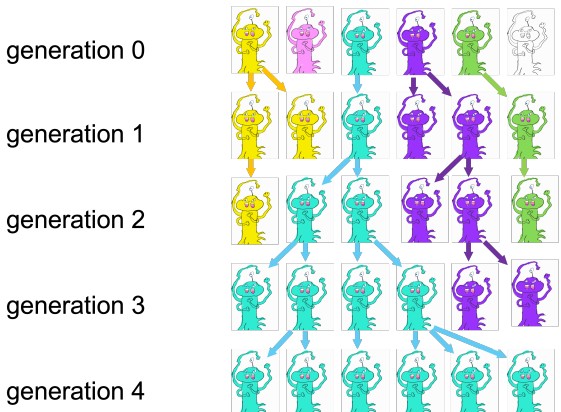

generation 0

generation 1

generation 2

generation 3

generation 4

Figure 7: Schematic illustration of the Wright-Fisher model, with alleles depicted as space aliens (graphics: Naomi Verhoek). Here, the population size $N = 6$. At each generation the population is obtained by random sampling with replacement from the previous generation. By generation 4 the cyan allele has fixed, i.e. it has taken over the entire population.

## 6.1 The Wright-Fisher model for genetic drift

The classic model for the effects of genetic drift on evolution is the **Wright-Fisher model** [21,22], named after the American geneticist Sewall Wright (1889–1988) and the British polymath Ronald Fisher (1890–1962). The Wright-Fisher model considers a population of fixed size $N$ that reproduces in discrete, non-overlapping generations. Considering for simplicity a haploid population (with one allele per individual), each new generation is created by simply sampling $N$ alleles at random (with replacement) from the current population. Figure 7 provides a schematic illustration of this process. At each generation, alleles can be stochastically lost from the population, so that ultimately one allele fixes, i.e. it takes over the entire population. Importantly, this is a neutral model – each allele has an equal chance to reproduce (and ultimately to take over the population), independently of all other alleles. The model is simple and generic. This simplicity is obtained by ignoring selection and mutation, along with almost all biological details; however the Wright-Fisher model can easily be extended to include selection by adding a bias when choosing alleles for the next generation, and mutation can be included by introducing new alleles at each generation [23].

Although it lacks biological realism, the Wright-Fisher model has the great advantage that it is easy to analyze mathematically. In each generation, the offspring are chosen according to a binomal process: $N$ "trials" are performed in which the chance of picking a particular allele is equal to its frequency (relative abundance) in the current population.

To be specific, let us suppose there are $M_t$ copies of a particular allele in the population (of size $N$), at time $t$. The frequency of the allele is then $X_t = M_t/N = x$. The number $M_{t+1}$ of copies of this allele in the next generation follows the binomal distribution with $N$ trials and probability $x$:

$$P[M_{t+1} = NX_{t+1} = k] = \binom{N}{k} x^k (1-x)^{N-k}, \tag{5}$$

where $\binom{N}{k} = N!/(k!(N-k!))$ is the binomial coefficient. We note that Eq. (5) depends only on the current state of the system (i.e. $X_t$). Therefore, the dynamics of the allele frequencies is Markovian, i.e. it is not history-dependent.

Eq. (5) allows us to calculate the mean and variance of the allele frequency $X_t = M_t/N$.[7]

---

[7]We use the fact that for a binomial distribution with parameters $N$ and $x$, as in Eq.(5), the mean is $Nx$ and the variance is $Nx(1-x)$.

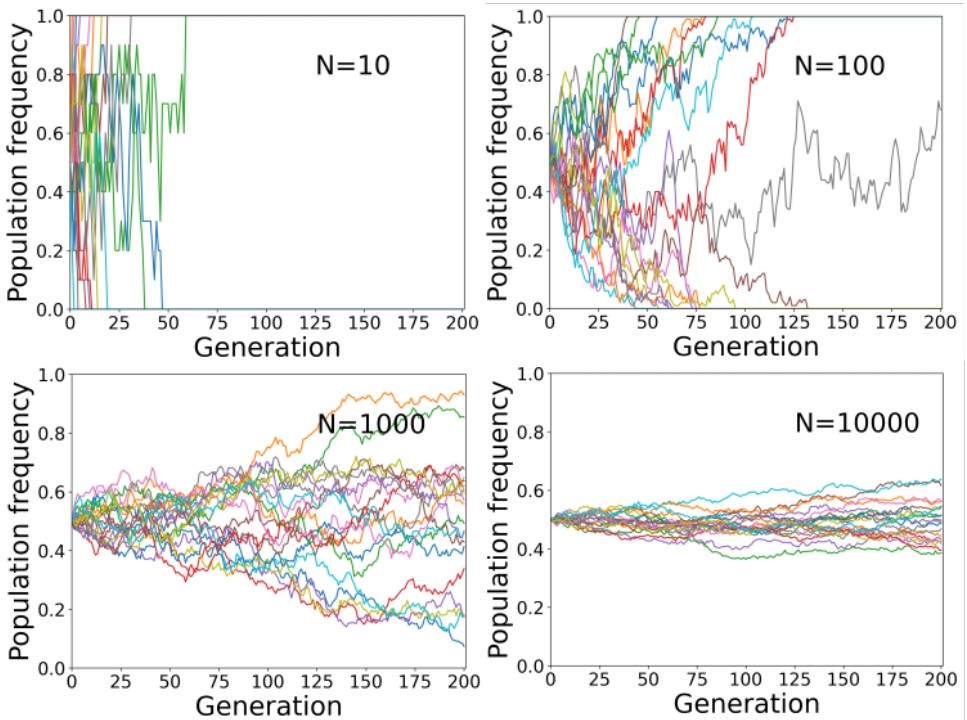

Figure 8: Simulations of the Wright-Fisher model for different population sizes $N$. The simulations are run for the case where there are 2 possible alleles, $A$ and $B$, which start at equal frequency. The coloured lines show replicate dynamical stochastic trajectories of the frequency of allele $A$.

The mean $E(X_t) = E(M_t)/N = x$, and the variance $V(X_t) = V(M_t)/N^2 = x(1-x)/N$. Since the variance in allele frequency scales as the inverse of population size $(1/N)$, stochastic fluctuations becomes more important as the population size decreases. This phenomenon is evident when we perform stochastic simulations of the Wright-Fisher model for different population sizes, as shown in Figure 8.

In the Wright-Fisher model, one allele ultimately takes over the population (as mentioned above). This is evident for the simulations of Figure 8, for the smallest value of the population size ($N = 10$) – here, all simulations end with fixation of one or other allele; thus, the frequency of allele $A$ becomes either 1 or 0 (with equal probability). We also observe some fixation events in the simulations for $N = 100$, but fixation does not happen on the timescale of the simulation runs for the larger population sizes shown in Figure 8.

Let us briefly think about allele fixation in the Wright-Fisher model. First, since the model is neutral, each individual allele copy that is present at the start has an equal chance of ultimately dominating the population. Therefore the probability $p_{\text{fix}}$ of ultimate fixation of a given allele is simply equal to its frequency: $p_{\text{fix}} = x$. This explains why we see roughly equal numbers of trajectories that end with $A$ fixing or becoming extinct for $N = 10$ in Figure 8, where $x = 0.5$ at the start. Next, the probability that an allele that has frequency $x$ will be lost in the next generation is the probability that it is not chosen in any of the $N$ trials associated with that generation – this is $(1-x)^N$. If $N$ is large, this probability will be vanishingly small, and we can expect fixation to take many generations. However if $N$ is small, $(1-x)^N$ is much larger, alleles are more often lost from the population and fixation occurs faster, consistent with the simulation results of Figure 8. Analytical calculation of the mean time to fixation in the Wright-Fisher model is possible via a diffusion approximation to the Wright-Fisher model [23], but will not be discussed in these notes.

## 6.2 Alternative models for genetic drift

The Wright-Fisher model is not the only model that is used to study genetic drift. One alternative is the Moran model, invented by Australian statistician Patrick Moran (1917–1988) [24]. Here, as in the Wright-Fisher model, the frequencies of alleles of different types are tracked. However, the dynamical rules differ from that of the Wright-Fisher model. In the Moran model, a single randomly selected allele is chosen to reproduce, while another randomly selected allele is chosen to die; doing this $N$ times corresponds to a generation. This is a birth-death process. An important difference between the Moran model and the Wright-Fisher model is that, in the Moran model, it is meaningful to think about continuous time, or overlapping generations, where at a given time, some individuals within the population have reproduced while others have not. In contrast, in the Wright-Fisher model, time is strictly divided into discrete generations, since the entire population is updated at once. This continuous time property of the Moran model is helpful for more advanced mathematical treatments such as mapping the allele frequency dynamics onto a diffusion equation, which allows quantities such as the mean time to allele fixation to be calculated. The models also differ in that in the Moran model, any individual only has zero or 1 offspring in the next generation, whereas in the Wright-Fisher model, an individual can, in principle, have up to $N$ offspring. The two models do, however, behave similarly in important ways. For example, in a 2-allele model that starts with a frequency $x$ of allele type 1, the probability of eventual fixation of this allele is $x$ in both models [25].

## 6.3 The rate of loss of genetic variation in the Wright-Fisher model

We have already seen that genetic drift leads to stochastic extinction of alleles and hence decreases the amount of genetic variation in the population. We now consider the loss of genetic variation using a different measure: the proportion of heterozygotes in a diploid population. To do this, we need to extend the Wright-Fisher model to account for a population of $N$ individuals, each of whom carry 2 alleles. As in section 4.4, we will assume that individuals mate at random and for simplicity we suppose that the individuals are monoecious (can be male or female).

To extend the Wright-Fisher model to the case of diploid individuals, we would in principle create an offspring individual by choosing two parent individuals and randomly selecting an allele from each parent. We would then repeat this procedure $N$ times to obtain $N$ offspring, each with 2 alleles. From the point of view of the allele pool this is almost, but not quite, the same as if we had simply sampled $2N$ alleles at random (with replacement) from the combined pool of $2N$ alleles in the current generation.[8] In this approximation, the diploid Wright-Fisher model is identical to the haploid model, but with $2N$ alleles instead of $N$.

Making this simplifying approximation, we define the **heterozygosity** $H$ as the probability that two alleles, chosen at random from the pool of $2N$ alleles, are different. We also define the quantity $G = 1 - H$, the probability that two alleles chosen at random are the same. Clearly, if every allele in the pool is the same then $G = 1$ and $H = 0$, whereas if every allele is different, $G = 0$ and $H = 1$.

We now calculate how the heterozygosity $H$ changes from generation to generation. To create the new generation, we sample $2N$ alleles at random with replacement from the existing allele pool. Let us suppose that the first offspring allele has been chosen. What is the chance that the second offspring allele will be the same as the first? There are two different ways that this could happen. Firstly, the second allele could have originated from the same parent as the first allele. Since all parent alleles have equal chance of being chosen, the probability

---

[8]The difference is that if we track individuals we would not allow an offspring to inherit both its alleles from the same parent, but this is allowed if we only track the allele pool.

that we chose the same parent again the second time is $P = 1/(2N)$. Alternatively, the second allele could have a different parent that is of the same type. The probability of this happening is $P = (1 - 1/2N)G$ (i.e. the probability $1 - 1/2N$ of having a different parent multiplied by the probability $G$ that the two parents are of the same type). This implies that the probability $G'$ that any two alleles in the offspring generation are the same is

$$G' = \frac{1}{2N} + \left(1 - \frac{1}{2N}\right)G\,. \tag{6}$$

We can now obtain an expression for the heterozygosity $H$ in the offspring generation:

$$H' = 1 - G' = H - \frac{H}{2N}\,,$$

where we have used the expression derived above for $G'$ and substituted in $H = 1 - G$. The change in heterozygosity per generation, $\Delta H = H' - H$, is then given by $\Delta H = -H/2N$, implying that

$$H(t) \simeq H(0)e^{-\left(\frac{t}{2N}\right)}\,, \tag{7}$$

where $t$ is time measured in generations. Therefore heterozygosity, which is a measure of genetic variation within the population, decays in time due to genetic drift, with a half-time of $2N \log 2$ generations.

To get a feeling for the practical implications of this result, let us consider a population of humans, consisting of 1 million individuals with a generation time of 20 years. Our calculation predicts that genetic drift will take 1.38 million generations, or 28 million years, to halve the heterozygosity of this population. This is roughly equal to the entire time since the evolution of primates on Earth. Therefore we can conclude that genetic drift does reduce genetic variation over time as alleles become extinct, but for large populations this is a very slow process. On short timescales, we can still expect to observe different genotypes in the relative abundances that are predicted by the Hardy-Weinberg equilibrium (section 4.4).

In reality, there are many reasons why we would not expect Eq. 7 to provide a quantitatively accurate prediction for the rate of loss of genetic variation in a human population over millions of years. A real population is different from the idealised Wright-Fisher model: among other things, some individuals fail to reproduce, individuals do not mate at random, and the population size can change drastically over time. To account for such effects in a coarse-grained way, it is common in population genetics to define an **effective population size** $N_{\text{eff}}$. This is defined as the population size in the Wright-Fisher model for which allele frequency dynamics (for example the rate of loss of genetic diversity (Eq. 7)) matches observations for a real population. $N_{\text{eff}}$ is generally less than the real population size [23, 26]. For example, the effective population size of the current human population has been estimated from DNA sequence variability to be of the order of only 10,000 [27].

## 6.4 Mutation-drift balance

What happens on long timescales, over which the loss of genetic variation due to genetic drift is relevant? On these timescales, the loss of alleles due to stochastic extinction is balanced by the creation of new alleles by mutation. This is known as **mutation-drift balance**.

To understand this, let us consider a Wright-Fisher model with mutations. We again consider a population of $2N$ alleles in which the offspring generation is created by sampling $2N$ alleles at random, with replacement, from the parent generation. However, now we add a new element: the offspring allele can mutate into a different type with probability $\mu$.

Repeating the analysis of section 6.3, let us again calculate the probability $G'$ that, having chosen one offspring allele, the next offspring is the same. Again, there are two ways that this

can happen: either the two offspring share the same parent, or they have different parents of the same type. However now there is the additional possibility that either of the offspring could have mutated to a new allele type, even if they were originally the same. The requirement that neither offspring allele has mutated introduces a factor $(1-\mu)^2$ into the expression for $G'$ compared to Eq. (6):

$$G' = \left[\frac{1}{2N} + G\left(1 - \frac{1}{2N}\right)\right](1-\mu)^2.$$

Assuming that the mutation rate $\mu$ is small, so that $(1-\mu)^2 \simeq 1-2\mu$, and neglecting terms in $\mu/N$, we arrive at

$$G' \approx \frac{1}{2N} + G - \frac{G}{2N} - 2G\mu.$$

Using $H = 1-G$ and $H' = 1-G'$ we can then express the heterozygosity $H'$ as

$$H' = H - \frac{H}{2N} + 2\mu(1-H),$$

so that the change in heterozygosity $\Delta H = H' - H$ from parent to offspring generations is given by

$$\Delta H = -\frac{H}{2N} + 2\mu(1-H). \tag{8}$$

Eq. (8) illustrates clearly the balance between loss of heterozygosity due to genetic drift (negative first term on the r.h.s.), and gain of heterozygosity due to mutations (positive second term on the r.h.s.). Setting $\Delta H = 0$ we find the equilibrium level of heterozygosity $H^*$:

$$H^* = \frac{4\mu N}{1 + 4\mu N}. \tag{9}$$

The fact that the mutation rate and population size occur only in the combination $4\mu N$ in Eq. (9) illustrates how genetic drift (which depends on $N$) and mutation have directly opposing effects on the level of genetic variation within the population.

## 6.5 The neutral theory of evolution

The concept that variation within a population can be explained by mutation-drift balance played a central part in a stochastic, view of evolution that emerged in the latter half of the 20th century, pioneered by the Japanese biologist Motoo Kimura (1924-1994). Kimura argued that at a molecular level, i.e. at the level of molecular changes in DNA or protein sequences, evolution is mostly driven by neutral genetic drift rather than by selection [28, 29].

Modern molecular techniques allow us to sequence genomes of multiple individuals from the same species rapidly and affordably. These techniques have revealed that individuals within the same species typically have many genetic differences, or **polymorphisms**. These polymorphisms accumulate from generation to generation due to mutation and most of them are neutral (have no effect on fitness). However in a finite population, genetic drift drives polymorphic variants to extinction (section 6.3), so from time to time a single variant **fixes**, i.e. takes over the entire population. Hence, molecular evolution can be viewed as a series of neutral mutation and fixation events. This view is illustrated schematically in Figure 9.

How often do we expect a new genetic variant to fix in a (diploid) population of size $N$? We can answer this question using a Wright-Fisher model with $2N$ alleles. If each allele mutates with probability $\mu$ per generation, the total new mutations per generation will be, on average, $2N\mu$. We know that in the Wright-Fisher model, each allele has an equal probability $1/(2N)$ of eventual fixation (section 6.1). Therefore the rate of fixation of new variants, per generation, is predicted to be $2N\mu \cdot (1/2N) = \mu$, independent of the population size.

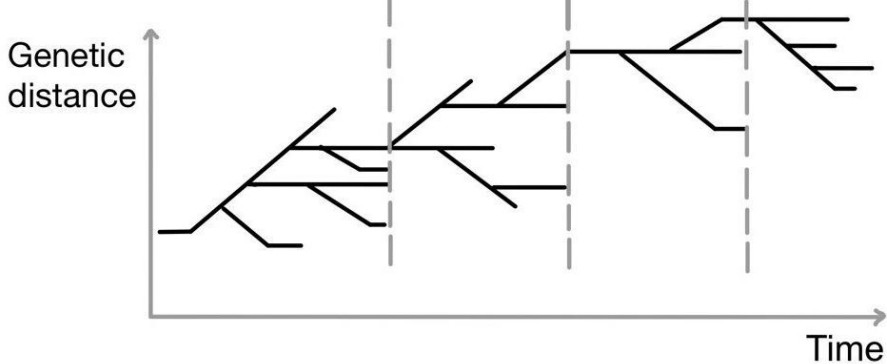

Figure 9: Illustration of the concept of molecular evolution as a series of neutral mutation and fixation events (graphic inspired by Ref. [30]). Neutral mutations happen often during the species' lineage, leading to polymorphic variation among individuals (illustrated here by branching lineages). As mutations accumulate over time, the lineage tree becomes more branched. However, from time to time a single variant fixes in the population and the other variant lineages become extinct (these fixation events are shown by the dashed grey lines). New polymorphism then accumulates starting from the lineage that has fixed.

Kimura tested this remarkable prediction by comparing the rate of amino-acid substitutions in the coding sequences for proteins during the evolutionary history of different vertebrate species, finding evidence that this rate is indeed conserved [28, 29].

Is this stochastic view of evolution in conflict with the Darwinian picture of natural selection, discussed earlier in these notes? No - both views are correct, and both types of evolution can be at play at the same time. Genetic drift is relevant for neutral and near-neutral mutations, that happen often at the molecular level, and for small population sizes. At the same time, Darwinian natural selection drives evolution in the case of non-neutral mutations that influence phenotype.

## 6.6 Lineage coalescence in the Wright-Fisher model

We conclude with a brief discussion of a different application of stochastic population genetics. In this approach, pioneered by John Kingman (born 1939), we look backwards in time. Starting with the genotypes of $n$ individuals in the present generation, we consider the lineage of ancestors going back in time from each individual, and we ask when these lineages coalesce. In other words, we would like to know the time since the **last common ancestor** of our chosen individuals.

As a simple example, we will calculate the time to the last common ancestor for two randomly selected alleles in the diploid Wright-Fisher model without mutations. We choose at random 2 of the $2N$ alleles in the population at the present time. We track back the lineage of each allele through the generations, and ask for the number $k$ of generations that we need to go back in time in order to find a common parent. In fact, since the Wright-Fisher model is stochastic, we aim to obtain the probability distribution for $k$.

Let us start by looking back one generation in time. The probability that our 2 chosen alleles share a common parent is $P(k = 1) = 1/(2N)$.[9] Therefore the probability that the 2 alleles do not share a parent is $1 - 1/(2N)$. In this case, we go a generation further back in time

---

[9]To see this, note that the parent of a given allele has an equal chance of being any of the $2N$ alleles in the previous generation, so that having identified the parent of the first allele, the probability that the second allele has the same parent as the first is $1/(2N)$.

and ask whether our 2 alleles share an ancestor 2 generations back. The probability of this is $P(k=2) = [1 - 1/(2N)] \times [1/(2N)]$.[10] The probability that there is no common ancestor either 1 or 2 generations back is $[1 - 1/(2N)] \times [1 - 1/(2N)] = (1 - 1/(2N))^2$. Continuing the pattern, we see that the probability that our alleles share an ancestor $k$ generations ago is

$$P(k) = \left(1 - \frac{1}{2N}\right)^{k-1} \left(\frac{1}{2N}\right). \tag{10}$$

Eq. (10) is actually a geometric distribution, for which we can easily find the mean $E(k)$:[11]

$$E(k) = \sum_{k=1}^{\infty} k P(k) = \left(\frac{1}{2N}\right) \sum_{k=1}^{\infty} k \left(1 - \frac{1}{2N}\right)^{k-1} = 2N. \tag{11}$$

Modelling the coalescence of lineages backwards in time is highly relevant in modern population genetics, since it allows one to infer the evolutionary history of species from genetic data of present-day (and sometimes past) individuals. Although we have illustrated the basic idea here, these calculations are actually much more complex, including not only mutation but also factors that have not been covered in these lectures, such as changes in population size, migration and genetic recombination [31].

---

**Summary of chapter 6**

- Stochastic processes in evolution are called genetic drift.

- The Wright-Fisher model is often used to study genetic drift.

- Genetic drift reduces variation in populations through stochastic extinction of variants, while mutations create new variation.

- At the molecular scale, mutation-drift balance often dominates over selection since many mutations are neutral or close to neutral.

- Stochastic population genetics can also predict the lineage history of populations.

---

## 7 Conclusion

These lecture notes provide a basic introduction to population genetic theory for a reader with a statistical physics background. The lectures were motivated by the observation that population genetics theory rarely features in biological physics courses, even though it forms the basis of our theoretical understanding of evolution. We are not experts in this topic and we apologize for any inadvertent errors. These notes certainly do not provide comprehensive coverage of this rich and complex topic. Not only are there many further interesting biological aspects (to mention just a few examples, genetic recombination, and spatial fragmentation of a population), but there is also a rich array of mathematical approaches that can be applied to this topic (for example, for the stochastic models, diffusion approximations and the use of master equation methods). For further reading, many excellent textbooks are available, as well as other lecture notes, for example Refs. [23, 26, 32–37].

---

[10]The first factor is the probability that there is no shared parent 1 generation back and the second factor is the probability that the parents of our alleles share a parent.

[11]In Eq. (11) we have used the fact that $(1-x)^{-2} = 1 + 2x + 3x^2 \ldots$

# Acknowledgments

The authors thank the organisers of the Les Houches School on Theoretical Biophysics and also thank all participants of the School for productive discussions that improved the content and presentation of the lectures. We are grateful to Naomi Verhoek for her graphical contributions to Figures 1, 3, 6 and 7.

**Funding information** RJA was supported by the European Research Council under Consolidator Grant 682237 EVOSTRUC and by the Excellence Cluster Balance of the Microverse (EXC 2051 - Project-ID 390713860) funded by the Deutsche Forschungsgemeinschaft (DFG). AI-R received funding from the European Union's Horizon 2020 research and innovation programme under the Marie Skłodowska-Curie grant agreement No 847718. SPL is funded by the LabEx DEEP (ANR-11-LABX-0044 and ANR-10-IDEX- 0001-02) and the Institut Curie. This publication reflects only the authors' views. The relevant funders are not responsible for any use that may be made of the information it contains.

**Author contributions** AI-R and SPL contributed equally to this work. RJA prepared and delivered the lectures. AI-R and SPL wrote the lecture notes. RJA, AI-R and SPL edited the lecture notes.

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
