# Peer review of "Population genetics: an introduction for physicists"

_SciPost Physics Lecture Notes, doi:SciPost Phys. Lect. Notes 89 (2024)_

## Round 3 · Referee Report · Anonymous (Referee 1) · 2024-9-4

Report

These lecture notes are extremely pedagogical, and explain key notions of population genetics without assuming prior biological or specialized knowledge. The explanations are clear and the derivations are presented in detail. Thus, I consider that the criteria for publication in this series of lecture notes are met.

Here are a few remarks and suggestions. 1- Natural selection is presented in the deterministic case, but the stochastic case centers on neutral mutations without selection (it is centered on the neutral Wright-Fisher model). While I understand that the time available for the lectures was limited, I believe that it would be a plus for readers to briefly mention that the Wright-Fisher model can be extended to the case with selection. Pointing to a reference where this can be found, and perhaps discussing the results in 1-2 sentences might also be good. 2- The stochastic part is treated within the Wright-Fisher framework, while the Moran model is mentioned in part 6.2. Adding a few words about what is different and what is similar between these two models might be a plus.

Recommendation

Ask for minor revision

  • validity: -
  • significance: -
  • originality: -
  • clarity: -
  • formatting: -
  • grammar: -

Author:  Rosalind Allen  on 2024-09-05  [id 4741]

(in reply to Report 1 on 2024-09-04)

We thank the referee for these very encouraging comments. We agree with both suggestions and we will make the suggested additions in our revised manuscript.

---

## Round 3 · Referee Report · Anonymous (Referee 2) · 2024-9-5

Report

These lecture notes are clear, well-written, and pleasant to read. The content is thoughtfully presented, which contributes to the overall readability and ease of understanding. Below are a few suggestions, which should help to clarify some points. I also noticed some typos that can easily be corrected, and are also listed below. Overall, the lecture notes meet the Journal's acceptance criteria, with their clarity, organization, and readability being strong points. The issues highlighted here are minor and can be easily addressed.

1- pp. 2-3: The sentences "in the late 20th century an opposing "dynamic" view emerged, which proposed that new species can arise, while old species can become extinct. An influential proponent of this idea was Georges Cuvier (1769-1832)" are slightly confusing regarding timing - did this view emerge in Cuvier's time or in the late 20th century? 2- p. 4, first line: "skin pigmentation or and eye color" should read "skin pigmentation or eye color". 3- p.5, section 3, 4th line from the end: hereditory -> hereditary. 4- p. 7 and Figure 2: It would be good to specify whether f(X) is assumed to be Gaussian, or approximated by a Gaussian, or neither of them and we are just focusing on its first two moments. If f(X) could in principle have any form, then it might be good to specify that the Gaussian in the figure is an example. 5- p. 15, Figure 5. The mutation is highlighted in red, but this is not very visible. Making it more visible, e.g. in bold, would facilitate understanding. Besides, it would be nice to also highlight the relevant codons (GAG and GUG), e.g. with a horizontal bracket. 6- p. 16, second-to-last paragraph: “However, this is a notoriously imprecise procedure, and not only because one relies on the mathematical model being the correct one. The mutant number distribution…”. These sentences could be simplified to facilitate understanding, e.g. as "However, this is a notoriously imprecise procedure. First, it relies on the mathematical model being correct, which may not be the case. Second, the mutant number distribution…" 7- p. 16, slightly below, "these mutants will multiple exponentially": multiple -> multiply. 8- p. 18, summary, bullet point 2, "the selection coefficient s = w − 1" should read "the selection coefficient is s = w − 1". 9- p. 18, summary, bullet point 3, "selection increase the proportion": increase -> increases. 10- p. 19, section 6.1. It might be helpful to say that selection/fitness can be incorporated when picking the next generation, and to add a reference about this more general case. 11- p. 19, section 6.1, just above the figure, "The model is simple and generic, but it ignores both selection and mutation, along with almost all biological details": It might be useful to reformulate this and to specify that generalizations exist and where to find them, e.g.: "The model is simple and generic. However, this simplicity is obtained by ignoring both selection and mutation, along with almost all biological details. Note that selection can be taken into account in the Wright-Fisher model [citation, e.g. Ewens' textbook], and mutation too." 12- p. 20, figure caption. It might be helpful to add "stochastic" before "trajectories". 13- p. 21, end of section 6.1, "A full analytical calculation of the mean time to fixation in the Wright-Fisher model is possible but will not be discussed in these notes.": it would be helpful to mention a reference where the mean time to fixation is treated (e.g. to Ewens' textbook). 14- p. 25, last sentence: it would be helpful to mention a reference where those topics are treated.

Recommendation

Ask for minor revision

  • validity: -
  • significance: -
  • originality: -
  • clarity: -
  • formatting: -
  • grammar: -

Author:  Rosalind Allen  on 2024-09-05  [id 4740]

(in reply to Report 2 on 2024-09-05)

We thank the referee for these very positive comments and for the helpful and detailed suggestions. We will certainly make the suggested changes in our revised version of the notes.

---

## Round 3 · Referee Report · Anonymous (Referee 3) · 2024-9-13

Report

The authors provide a clear introduction to the topic, striking a nice balance between detailed mathematical derivation and biological insight. I believe the lecture notes meet the criteria for publication, but I do have a few questions and comments:

  • Given its historical significance, I would have enjoyed reading more about the Luria-Delbruck experiments. Even a short, illustrative figure to demonstrate the key ideas behind the experiment would benefit the discussion. As it is currently written, the authors devote a somewhat significant amount of text to the experiment, but forgo mentioning the necessary details to follow closely. Also, it was my understanding that the experiment's significance in part stems from the observation that mutations can arise in the absence of selective forces. It would be helpful to mention this in the text.

  • When discussing alternatives to Wright-Fisher, it would be good to mention terms and provide references to some continuous approaches (stochastic calculus, master equations, diffusion approximations, etc.). The interested reader can follow these references, and I assume that many of the tools will be familiar to physicists.

  • This is a minor suggestion, but if it can be fit into the text naturally I think the notes would benefit from a discussion of the difference between the true population size and the effective population size. When comparing predictions from population genetics models to real data, this distinction can be important. The simplest example that comes to my mind is the decay of heterozygosity when the population size changes in each generation.

Some typos I found

-"siye" should be "size" on page 24

Recommendation

Ask for minor revision

---

## Editorial Decision

published